# Reduction Temperature-Dependent Nanoscale Morphological Transformation and Electrical Conductivity of Silicate Glass Microchannel Plate

**DOI:** 10.3390/ma12071183

**Published:** 2019-04-11

**Authors:** Hua Cai, Yong Sun, Xian Zhang, Lei Zhang, Hui Liu, Qing Li, Tiezhu Bo, Dongzhan Zhou, Chen Wang, Jiao Lian

**Affiliations:** Key Laboratory of China Building Materials Industry for Special Photoelectric Materials, Institute of Special Glass Fiber and Optoelectronic Functional Materials, China Building Materials Academy, Beijing 100024, China; mutong1007@163.com (Y.S.); zhangxian@163.com (X.Z.); zhanglei@163.com (L.Z.); 13910289674@139.com (H.L.); liqing13_ren@163.com (Q.L.); botiezhu0318@163.com (T.B.); zhoudongzh668@163.com (D.Z.); wangchen@cbmamail.com.cn (C.W.); lianjiao630@163.com (J.L.)

**Keywords:** microchannel plate, microscopic potential distribution, Kelvin force microscopy, nanoscale morphological transformation, bulk resistance

## Abstract

Lead silicate glasses are fundamental materials to a microchannel plate (MCP), which is a two dimensional array of a microscopic channel charge particle multiplier. Hydrogen reduction is the core stage to determine the electrical conductivity of lead silicate glass MCP multipliers. The nanoscale morphologies and microscopic potential distributions of silicate glass at different reduction temperatures were investigated via atomic force microscope (AFM) and Kelvin force microscopy (KFM). We found that the bulk resistance of MCPs ballooned exponentially with the spacing of conducting islands. Moreover, bulk resistance and the spacing of conducting islands both have the BiDoseResp trend dependence on the hydrogen reduction temperature. Elements composition and valence states of lead silicate glass were characterized by X-ray photoelectron spectroscopy (XPS). The results indicated that the conducting island was an assemblage of the Pb atom originated from the reduction of Pb^2+^ and Pb^4+^. Thus, this showed the important influence of the hydrogen temperature and nanoscale morphological transformation on modulating the physical effects of MCPs, and opened up new possibilities to characterize the nanoscale electronic performance of multiphase silicate glass.

## 1. Introduction

Lead silicate glass is particularly useful for the manufacture of a microchannel plate (MCP), which is a special designed two dimensional array of microscopic channels for charge particles multiplied, has been widely used in astronomical, biological imaging, nuclear science and weak signal detection applications [1,2,3,4,5,6,7,8]. The core and clad lead silicate glasses formed the microchannel structure after drawing into fibers, assembling to bundle, fusing, slicing and polishing to wafers and wet chemically etching to remove the cores. Then after a hot hydrogen reduction it produced a conducting layer and formed a secondary electron emission layer on the surface of the microchannel wall, lastly, a nichrome layer was evaporated to provide for electrical contact. The hydrogen reduction was one of the most important processes where it possessed proper electrical conductivity. This fact and technical feasibility has increased exploratory interest in these glasses.

The bulk resistance of any microchannel plate can determine the application field of the MCP multiplier, which influences the dynamic range and voltage tolerance of MCP. The requirements of the dynamic range and voltage tolerance for the bulk resistance are the exact opposite. Dynamic range is defined as the range of the operational linear relationship between input and output currents of the MCP multiplier under a fixed external voltage. Voltage tolerance means the maximum working voltage that the MCP multiplier can be endured. As a key electron multiplier material, lead silicate glass MCP is required to have a good dynamic range and voltage tolerance for better applications. Existing literature studies have shown that reducing the bulk resistance benefits to extend the dynamic range [9,10], while increases the Joule heat production [11], which might lead to fatal defects and MCP multiplier malfunction. Thus, a suitable bulk resistance is the key to high performance and stability of the microchannel plate. For better applications, therefore, it is important to know exactly how to adjust the bulk resistance to the desired value to establish the relative maximum operational electron multiplication dynamic range and best voltage tolerance.

The electrical conductivity (the bulk resistance) of MCP can be modulated by the composition of glass materials and the hydrogen reduction parameters. The effects of the composition of glass materials on the conductivity of MCP will be discussed in detail in a subsequent paper. In this paper, we focused on the effect of the reduction procedure on the bulk resistance of MCP. Hydrogen reduction is the key procedure to achieve considerable electrical conductivity for electron multiplication operation [12]. Heretofore, many researchers note that lead oxides in silicate glass would be reduced into lead by hot hydrogen, then form the conducting layer in the reduction process, substantially decreased the resistivity of glass [11,12,13]. Some investigations concluded that the depth distribution of the Pb ion [14] or surface topography [15] of reduced lead/bismuth silicate glass both showed a reduction in temperature-dependent properties, which influenced the entire surface conductivity of the channeltron. Our previous work also showed that the reduction temperature would change the morphologies [16] and chemical states [17] of the channel surface in the MCP multiplier with the analysis of the atomic force microscope (AFM) and X-ray photoelectron spectroscopy (XPS), respectively. However, hardly any systematic research has been done to understand the performance variation laws of the nanoscale microstructures and surface micro-conductivity in lead silicate glass as a linear change of reduction temperatures.

As powerful tools, AFM and Kelvin force microscopy (KFM) have been widely utilized for inspecting nanoscale conductor/semiconductor/insulator materials [18,19,20]. KFM has a super-high resolution and local electric properties probing ability based on a tip-bias, which make it a powerful method of investigating surface charge/potential difference distributions at the nanoscale.

In this paper, we used AFM, KFM and XPS to investigate the local morphological transformation, potential distributions and elemental valence states of lead silicate glasses of channel wall surface in MCPs at the linear changed hydrogen reduction temperatures, respectively. Meanwhile, we measured the bulk resistances of MCPs under the corresponding conditions in order to establish the relationship among nanoscale microstructure, surface microscopic charge/potential distributions and macroscopic electrical properties. It can be found that the hydrogen reduction temperature plays a critical role in the nanoscale morphologies and the related electrical properties.

## 2. Materials and Methods

MCP samples were provided by the MCP Research Center of China Building Materials Academy (Beijing, China). The nominal composition of CML865 lead silicate glass is shown in Table 1. SiO_2_ was quartz sand with high purity; other materials were chemical grade purity. After being mixed, raw materials were decanted into a quartz crucible, melted in an electric furnace under atmospheric conditions and clarified at 1800 K in sequence. A glass tube with a diameter of 30 mm and thickness of 3.0 mm was then obtained by casting at 1550 K, and annealed at 650 K.

Glass rod and tube were assembled into glass-core-clad, drawn to a small diameter, fused together into a bundle, sliced and polished into wafers with a diameter of 25 mm and a thickness of 0.30 mm, etched away the cores to form the microchannel structure with a 6 µm channel diameter and heat treated in a hydrogen atmosphere (1.1 × 10^5^ Pa) in the reaction chamber to produce a semiconducting layer. The hydrogen reduction parameters are shown in Table 2. Then nickel-chromium electrodes were vacuum-deposited on both faces of the microchannel plate by a Buhler SYRUS 900 evaporation system (Bavaria, Germany). Therewith, samples were vacuum-packaged immediately until testing.

Bulk resistances of MCPs measured by the Vacuum Photoelectron Test Facility (VPTF, Nanjing, China), as shown in Figure 1, the DC voltage *V_D_* applied across both ends of the MCP multiplier and the strip current *I_S_* was the sum of all currents flowing through every single parallel channel wall of the MCP. We measured *V_D_* and *I_S_* by using a voltmeter and nano-ammeter, respectively. Meanwhile, we maintained voltage *V_D_* at 500 V. So, the value of bulk resistance equals the applied voltage *V_D_* divided by strip current *I_S_*.

The sample sketch for AFM, KFM and XPS testing is shown in Figure 2. Samples were set into a 100-grade purification table, sliced by a diamond cutter and analyzed the microchannel inner surface of longitudinal sections in the MCP. 

In the air ambient, a KFM based on AFM (model SPM-9700, Shimadzu, Kyoto, Japan) was used to analyze the surface morphology and the potential distribution of the MCP channel wall in a dynamic non-contact mode with the 0.01 nm Z-axis resolution, a silicon electrical measure probe with a platinum coating and reflex side aluminum coating (OMCL-AC240TM-B3) and 1 µm × 1 µm scanning area. Nova P8 software (2010 NT-MDT 1.1.0 version, Moscow, Russia) was used to calculate the statistic distribution of phases on the surface of channel walls. The measurements by XPS (model AXIS Supra, Kratos Analytical, Manchester, UK) were performed with Al Kα radiation (*hν* = 1486.6 eV). The pressure in the measurement chamber was about 2 × 10^−9^ torr and the testing temperature was about 18 °C. The XPS spectra were collected with an emission current of 15 mA, pass energy of 80 eV and sweep time of 60 s. The step size of 0.1 eVwas employed and each peak was scanned twice. The C1s binding energy (BE) of 285 eV was used as a calibration value.

## 3. Results and Discussion

### 3.1. Bulk Resistances at Different Reduction Temperatures

The bulk resistances of the CML865 lead silicate glass MCPs at reduction temperature controlled from 620 K to 770 K were shown in Figure 3 (the solid line), which firstly decreased dramatically and then increased tardily and obtained the minimum between 670 K and 690 K. At the 95% confidence level, the bulk resistance-reduction temperature curve (R-T_r_ curve) was fitted with the BiDoseResp function (a non-linear two-step regression analysis) in the Levenberg-Marquardt iteration algorithm by using OriginLab software (V 7.0, Northampton, MA, USA), as shown in Figure 3, the red dash line).

In this research, the coefficient of determination (R^2^) of the fitted R-T_r_ curve was about 1, meaning the goodness-of-fit was great; the relationship between R and T_r_ was best approximated by the BiDoseResp distribution under the condition of 95% confidence. According to the fitting eigenvalues of R-T_r_, the bulk resistance showed a minimum of 6.88 MΩ at 685 K, but fluctuated drastically in the range of 620–660 K, almost not changed in the range of 670–700 K, then gradually increased to the range of 700–770 K. The fitting curve equation between the bulk resistance and reduction temperature was as follows:(1)$$R=6.78+6712.72\left[\frac{0.993}{1+10^{\left(624.75-T_{r}\right)\left(-0.1\right)}}+\frac{0.007}{1+10^{\left(754.41-T_{r}\right)\left(0.04\right)}}\right],R2=0.9999$$

In the range of 620–685 K, the slope of the bulk resistance decrease was about −0.1, while it was about 0.04 in the range of 685–770 K. Meanwhile, the proportion of the bulk resistance decreased and increased with increasing temperature and was about 99.3% and 0.7%, respectively. The bulk resistance of MCP is mainly dependent on the physicochemical properties of the conduction phase in silicate glass [11,12,16,17]. This sigmoidal curve indicated the presence of two distinct equilibriums at the temperature for conductive phases. Therewithal, we probed the nanoscale morphological transformations and the microscopic potential distributions of lead silicate glass on the surface of the channel wall of MCPs at different hydrogen reduction temperatures.

### 3.2. XPS Spectra of Pb

Figure 4 represented the XPS spectra of the Pb element of lead silicate glasses surface before and after reduction. The Pb ion was kept in a divalent state or a tetravalent state before reduction due to the interaction of Pb–O or O=Pb=O (Figure 4a), while some shoulder peaks of Pb4f occurred in reduced samples (Figure 4b–f). After decomposing Pb 4f 7/2 into three peaks, the binding energy of 138.5–138.6 eV was associated with a network modifier of Pb^2+^ 4f 7/2, 137.7–138.2 eV was associated with a network modifier of Pb^4+^ 4f 7/2 and 136.6–136.8 eV should be attributed to Pb^0^ 4f 7/2 in the reduced sample. Some Pb^2+^ or Pb^4+^ ions were reduced into Pb. Pb^2+^, Pb^4+^ and Pb^0^ coexisted on the surface of reduced lead silicate glasses. With the hydrogen reduction and the rising reduction temperature, the binding energies of Pb^4+^ 4f, Pb^2+^ 4f and Pb^0^ 4f peaks resulted in a tiny negative shift. We believed that the lower electronegativity of Pb^0^ led to an increasing electron density and these decreasing binding energies.

It is worth noting that the peak area ratio of Pb^4+^ 4f continued to decline by raising the reduction temperature (Figure 5, the gray circles), even it was negligible at 770 K, shown in Figure 4e and Table 3. The MCP glass was generally reduced at 690–770 K for the applicable bulk resistance, the concentrations of Pb^4+^ 4f 7/2 were almost less than 4% (4.2–1.3%, as shown in Figure 4e,f), so that might account for the neglect of the PbO_2_ in many investigations into XPS spectra of the reduced lead silicate glass [16,17,21].

Meanwhile, as the hydrogen temperature increased, the concentration of Pb^0^, PbO and PbO_2_ showed different changing tendency. The concentration of Pb^0^ 4f 7/2 first increased and then reached a level (about 16.7%) from 620 K to 770 K, shown in Figure 5 (the purple squares); the content of Pb^4+^ 4f 7/2 showed a progressive decreasing with increasing reduction temperature, the decreasing amplitude of which was great in the range of 640 K to 690 K (almost −140%), while a smaller amplitude was in the range of 620–640 K or 690–770 K (almost −60%); the lever of Pb^2+^ 4f 7/2 was volatile from 32.3% to 38.9% due to the rising reduction temperature, as shown in Figure 5 (the blue dots). The concentrations of PbO and Pb^0^ were not in the inversely proportional relationship, which confirmed that the other compounds enabled to react to Pb^0^ in the silicate glass, such as PbO_2_. These non-linear rising/declining Pb concentration-hydrogen temperature behavior was attributed to the behaviors of thermodynamics and kinetics of PbO and PbO_2_ in the hydrogen reduction of lead silicate glass MCPs, which will be discussed in detail in Section 3.4. 

Moreover, we noticed that the concentration of Pb^0^ 4f 7/2 was kept a constant of about 16.7% at 690 K and 770 K, where the bulk resistance of MCPs was 6.5 MΩ and 50 MΩ, respectively; the two had seven times difference. It indicated that the bulk resistance of MCP was not just relevant to the content of Pb^0^ in the conducting layer of the channel wall, which might also be related to the morphology of conductive phases. To understand this seemingly paradoxical property, we measured the surface morphology and microscopic potential distribution of lead silicate glass MCPs by AFM and KFM.

### 3.3. Morphology Evolution of Pb Aggregates

The typical surface morphology and microscopic potential distribution of lead silicate glass MCPs was detected by AFM and KFM, respectively, as in Figure 6. KFM is based on the non-contact AFM (a dynamical mode) and minimizes the electrostatic interaction between the scanning tip and the surface. We used a frequency modulation mode, in which the electrostatic force gradient is minimized by the application of a DC bias voltage at 1.0 V. The coated highly n-doped silicon tip with an apex of about 15 nm and a resonant frequency of 70 kHz. The surface potential was calibrated by a stainless steel wafer before the experiment. The differences between the upper and the lower ranges of surface potentials were set into ~300 mV in the KFM images. For semiconductors, the contact potential difference is determined, which is related to the sample’s work function, while for insulators information about local charges is obtained [23].In this study, the bright granular phase occurred on the surface and diffused in the dark glass matrix in the AFM mode, correspondingly, a blue phase (with a lower potential) occurred on the surface and diffused in the red glass matrix (with a higher potential) in KFM mappings. 

The distribution of contact potential differences (CPD) of reduced lead silicate glasses at the reduction temperatures from 620 K to 770 K was shown in Figure 7. The extrema of CPD (Figure 7h) at different reduction temperatures were investigated with SPM-9700 KFM-analysis software, the averages of CPDs increased slowly (from −200 mV to −740 mV) after an initial decrease (from −660 mV to −515 mV). The difference between the maxima and the minima of CPDs was about 460 mV (the largest) at 690 K, where the bulk resistance was also the lowest.

The spacing of conducting islands was analyzed by a grain analysis module in the Nova P8 software with the KFM topographies. After screening for the CPDs and the areas for particles (the threshold of CPDs was calculated by the software automatically according to the Otsu’s method, the areas of particles were set as 100 nm^2^), the numbers (N) of conducting islands on 1 µm × 1 µm images were obtained. The average distance (D) between conductive islands was calculated as follows:(2)$$D=\frac{1000^{2}}{N}$$

The curves of the average distances of conducting islands and the bulk resistance of MCPs with the reduction temperature were shown in Figure 6. In which, solid lines represented the measurement data and dotted lines were the nonlinear fitting data. At the 95% confidence level, the spacing-reduction temperature curve (D–T_r_ curve) was fitted with the BiDoseResp function in the Levenberg-Marquardt iteration algorithm (as shown in Figure 8, the blue dash line). Bulk resistance and conducting island spacing both had the BiDoseResp dependence on the hydrogen reduction temperature. That is, the conductive characteristics of the microchannel plate were closely related to the spacing of conducting nano-islands in lead silicate glass.

In this research, the R^2^ for the fitted D–T_r_ curve was about 0.99. According to the fitting eigenvalues of D–T_r_, the spacing showed a minimum of 171 nm at 685 K, but fluctuated drastically in the range of 620–660 K, almost not changed in the range of 670–700 K, then gradually increased in the range of 700–770 K, resembling that of the R-T_r_ curve. The fitting curve equation between conducting islands’ spacing and the reduction temperature was as follows:(3)$$D=176.66+1355.18\left[\frac{0.899}{1+10^{\left(604.95-T_{r}\right)\left(-0.035\right)}}+\frac{0.101}{1+10^{\left(758.59-T_{r}\right)\left(0.026\right)}}\right],\;R^{2}=0.999$$

The slope of the bulk resistance was about −0.035 and 0.026 in the range of 620–685 K and 685–770 K, respectively. Meanwhile, the proportion of the bulk resistance decreased and increased with the increasing of temperature and was about 90% and 10%, respectively. This sigmoidal curve indicated the presence of two distinct equilibriums at temperature either lower or higher than the fully aggregation and growth stages value of the conductive phase. It was suggested that the new generated granular in the reduced lead silicate glasses of MCPs should be an aggregate of Pb atoms with the synthesis analysis by XPS spectra, AFM topographies and KFM images.

### 3.4. Thermodynamic and Kinetic of the Reducing Pb

It was assumed that hydrogen diffuses into the sample bulk and interacts with PbO/PbO_2_ with the formation of metal lead. The lead silicate glass might undergo the following reactions to the thermal hydrogen reduction (620–770 K):(4)$$A:\;\mathrm{PbO}\left(s\right)+H_{2}\left(g\right)=\mathrm{Pb}\left(l\right)+H_{2}O\left(g\right)\;\Delta G_{1}$$
(5)$$B:\;\mathrm{PbO}\left(s\right)+H_{2}\left(g\right)=\mathrm{Pb}\left(g\right)+H_{2}O\left(g\right)\;\Delta G_{2}$$
(6)$$C:\;{\mathrm{PbO}}_{2}\left(s\right)+2H_{2}\left(g\right)=\mathrm{Pb}\left(l\right)+2H_{2}O\left(g\right)\;\Delta G_{3}$$
(7)$$D:\;{\mathrm{PbO}}_{2}\left(s\right)+2H_{2}\left(g\right)=\mathrm{Pb}\left(g\right)+2H_{2}O\left(g\right)\;\Delta G_{4}$$
(8)$$E:\;2{\mathrm{PbO}}_{2}\left(s\right)=2\mathrm{PbO}\left(s\right)+O_{2}\left(g\right)\;\Delta G_{5}$$
where s is the solid state, g is the gaseous state and l is the liquid state. According to the theory of reaction thermodynamics, a negative ΔG is a necessary condition for the spontaneity of processes at constant pressure and temperature. The Gibbs free energy changes (ΔG) of the above five reactions with the reduction temperatures were calculated using the HSC Chemistry thermodynamic software (6.0 version, Helsinki, Finland), as shown in Figure 9. It showed that the reactions of A,C and D (ΔG_1_, ΔG_3_, ΔG_4_ > 0)were spontaneous but the reaction of B was non-spontaneous (ΔG_2_ > 0) in the temperature range of 620–770 K. A critical temperature zone of the non-spontaneous/spontaneous for the reaction of *E* was about 670–690 K (when T = 690–770 K, ΔG_5_ > 0). 

Above all, we speculated that the D-T_r_ and R-T_r_ curves were closely related to the thermodynamics and kinetics of the reducing lead, and the nucleation and growth behaviors of [Pb]-islands. We divided the reduction temperature into three ranges as shown in Figure 8. In the first stage (620–670 K), the lead oxides (PbO, PbO_2_) in the lead silicate glass were reduced into Pb (liquid or gas) continuously in the thermal hydrogen. The reaction rates of A, C and D increased with the raising reduction temperature, thus, a greater content of Pb was obtained at a higher reduction temperature in the same reducing time, which slumped the spacing of [Pb]-islands and the bulk resistance of MCPs. The Pb(g) could be out of the reaction furnace via the flowing hydrogen, which accelerated the reduction reactions of lead oxides. However, the second stage (670–690 K) was just in the critical temperature zone of reaction *E*, in which, some PbO_2_ could be thermal decomposed into PbO, according to the calculated Gibbs free energy changes, the total content of Pb(g) in the reduction reaction might be decreased, while Pb(l) was diametrically opposed. Therefore, the changes of the reaction rate of PbO_2_/PbO with temperature were lower in the second stage than that in the first stage, and formed a new type of equilibrium of Pb nucleation and growth. In the third stage (690–770 K), the required reduction time to reach the same content of Pb was shortened greatly when the temperature exceeded 690 K, as shown in Figure 10 [24], so the generated Pb(l) would increase. With the increasing temperature, the driving force of the migration and convergence of Pb(l) droplets was enhanced, entering to the agglomeration and growth steps of [Pb]-islands. The higher the reduction temperature was, the shorter the complete reduction time ([Pb]-islands nucleation time) of lead oxides, thus, the longer growth time of [Pb]-islands in the same reduction time, inducing a greater separation distance between [Pb]-islands. The resistance increased exponentially with an increasing hopping distance in the electronic hopping conduction mechanism [25]. So, the bigger spacing of [Pb]-islands, the farther the hopping distance of electron under the applied voltage and a bigger bulk resistance of MCPs. Therefore, the bulk resistance of MCPs and the spacing of [Pb]-islands both have the BiDoseResp trended dependence on hydrogen reduction temperature in the whole three stages.

## 4. Conclusions

The lead silicate glass employed in this investigation was on the surface of the microchannel wall of MCPs reduced by hydrogen with a constant pressure, flow and reduction time at the reduction temperature from 620 K to 770 K.

AFM, KFM and XPS detections indicated that aggregates of the Pb atom derived from the reduction of Pb^2+^ or Pb^4+^ dispersed on the working surface of the microchannel. The bulk resistance of MCPs was not only related to the content of the Pb atom in the conducting layer of the microchannel wall, but was also relevant to the morphology of conductive phases. We found that the bulk resistance of MCPs ballooned exponentially with the spacing of [Pb]-islands. It showed that the bulk resistance of MCPs and the spacing of [Pb]-islands both had the BiDoseResp trend dependence on the hydrogen reduction temperature in the range of 620–770 K. These sigmoidal curves indicated the presence of two distinct equilibriums at the temperature for the conductive phases, which were closely related to the thermodynamics and kinetics of the reducing lead and the nucleation and growth behaviors of [Pb]-islands. In the lower reduction temperature stage (620–670 K), the nucleation of [Pb]-islands dominated. In the second stage (670–690 K), some PbO_2_ could be thermal decomposed into PbO, the total content of Pb(g) in the reduction reaction might be decreased, while Pb(l) was diametrically opposed. Therefore, a new type of equilibrium of Pb nucleation and growth was formed. In the third stage (690–770 K), the growth of [Pb]-islands dominated. 

Besides, the analyses of surface topographies and extrema of contact potential differences (CPD) of reduced lead silicate glasses with KFM indicated that the bright granular phase occurred on the surface and diffused in the dark glass matrix in the AFM mode, correspondingly, a blue phase (with a lower potential) occurred on the surface and diffused in the red glass matrix (with a higher potential) in the KFM mappings. It opened up a new possibility to characterize the nanoscale electronic performance of multiphase silicate glass.

## Figures and Tables

**Figure 1 materials-12-01183-f001:**
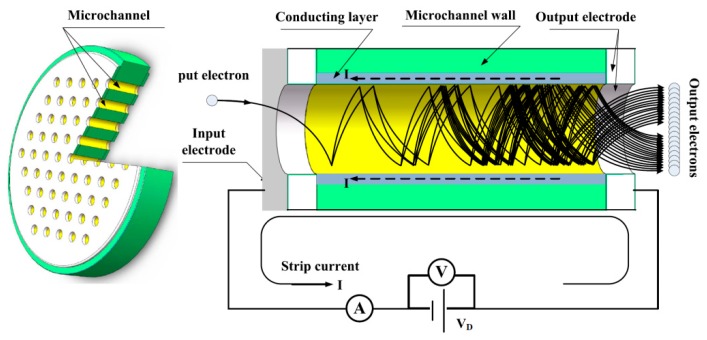
The schematic structure of the microchannel plate (MCP) and electron avalanche in a single channel. An incident electron strikes the input surface of a microchannel and produces one or more secondary electrons. The voltage *V_D_* applies across both ends of the MCP, the strip current *I_s_* follows through the conducting layer in the channel wall and supplies the current released from it.

**Figure 2 materials-12-01183-f002:**
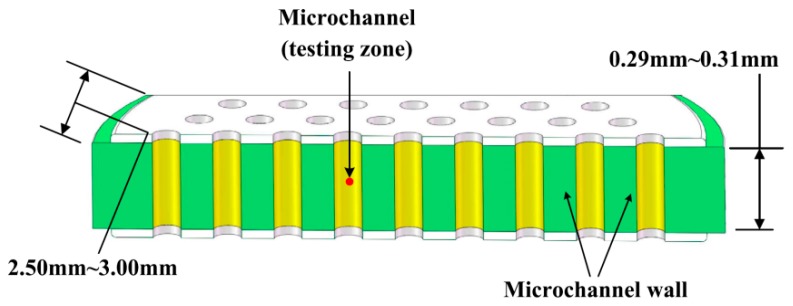
The sample sketch for properties characterized by atomic force microscope (AFM), Kelvin force microscopy (KFM) and X-ray photoelectron spectroscopy (XPS). AFM and KFM measurements were both in a low frequency vibration isolation device to reduce the external disturbance. XPS experimentation was carried out under ultra high vacuum (≤2 × 10^−9^ torr) with Al Kα radiation (*hν* = 1486.6 eV), the detected binding energies were from 0 eV to 1350 eV. The temperatures of samples were all no more than 20 °C in these three tests.

**Figure 3 materials-12-01183-f003:**
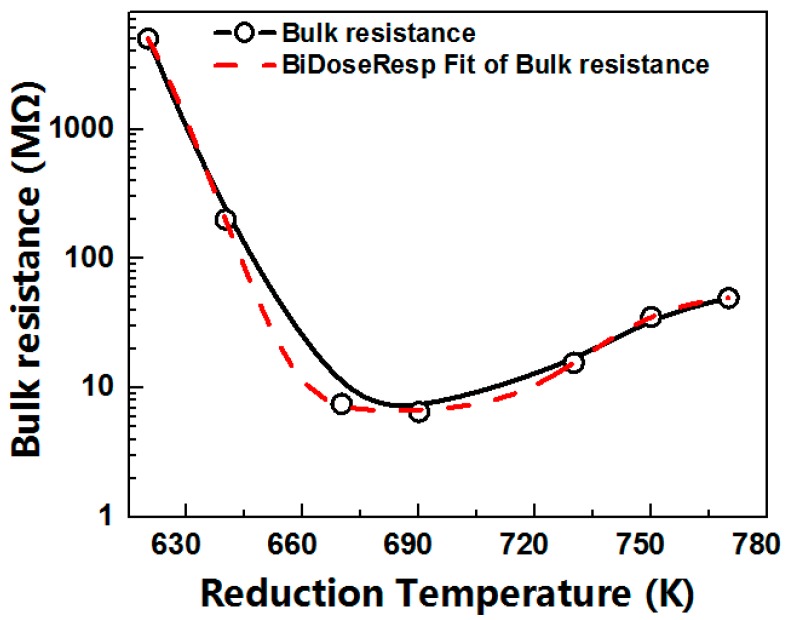
The bulk resistances of the CML865 lead silicate glass MCPs at the reduction temperature controlled from 620 K to 770 K. As the reduction temperature linear increased, the bulk resistances decreased dramatically firstly and increased tardily secondly, showing a BiDoseResp trended bulk resistance-reduction temperature behavior.

**Figure 4 materials-12-01183-f004:**
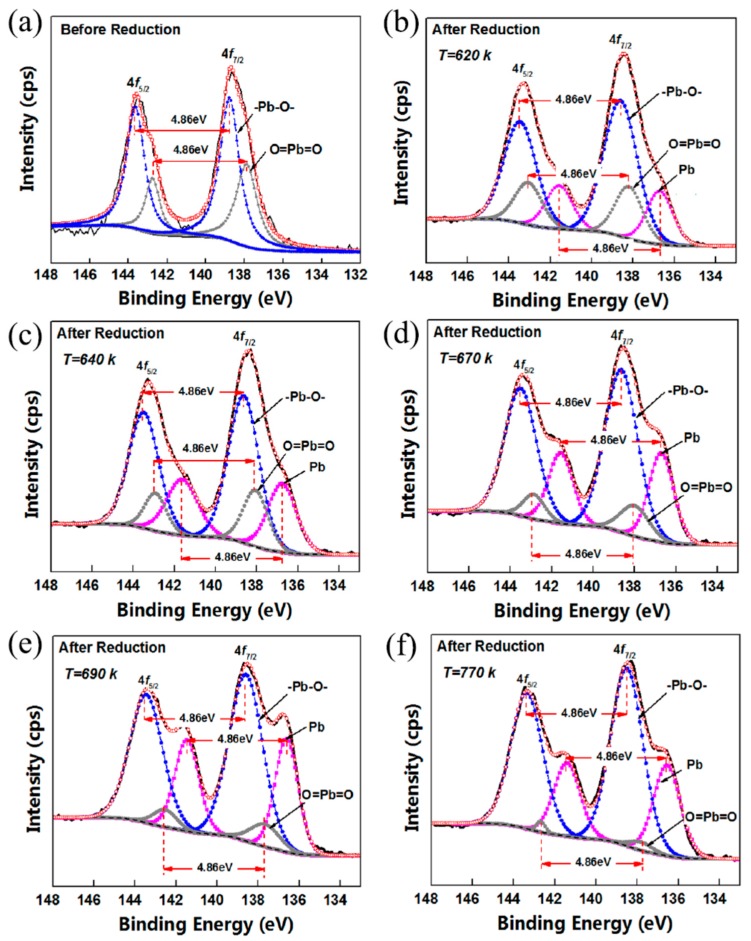
XPS spectra of Pb4f before and after reduction. (**a**) Before reduction; (**b**–**f**) after reduction; (**b**) T_r_ = 620 K, (**c**) 640 K, (**d**) 670 K, (**e**) 690 K and (**f**) 770 K, respectively; the binding energy differences between Pb 4f 7/2 and Pb 4f 5/2 of Pb^0^, Pb–O and O=Pb=O were set as 4.86 eV in the peak-fit processing according to the ”handbook of X-ray photoelectron spectroscopy” [22], the [Pb] ”shoulder” peaks were observed in the XPS spectrum after reduction.

**Figure 5 materials-12-01183-f005:**
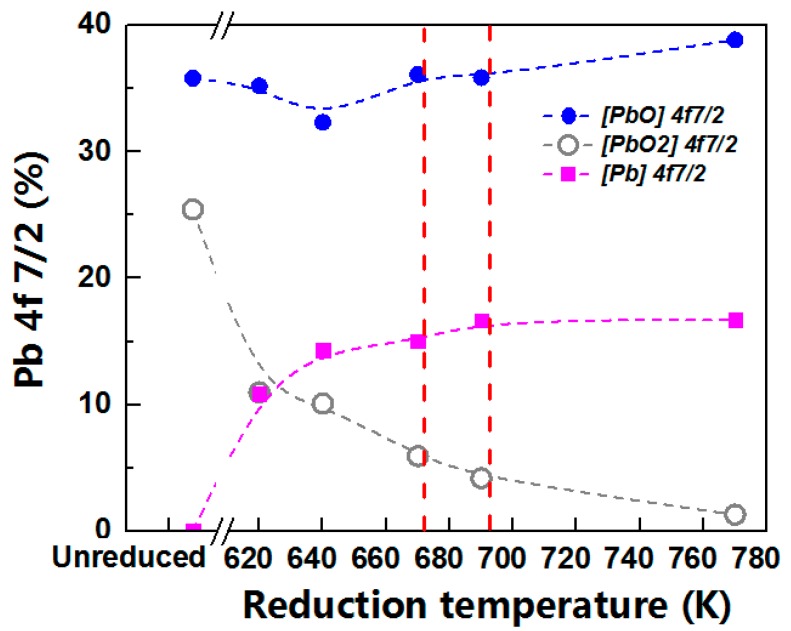
The concentration of Pb 4f 7/2 in the conducting layer of the channel wall before and after hydrogen reduction. On the left of the break in abscissa was before reduction, on the left of which was after reduction at different reduction temperatures. The purple squares, blue dots and gray circles means the area ratio of 4f 7/2 binding energy peak of Pb^0^, PbO and PbO_2_, respectively.

**Figure 6 materials-12-01183-f006:**
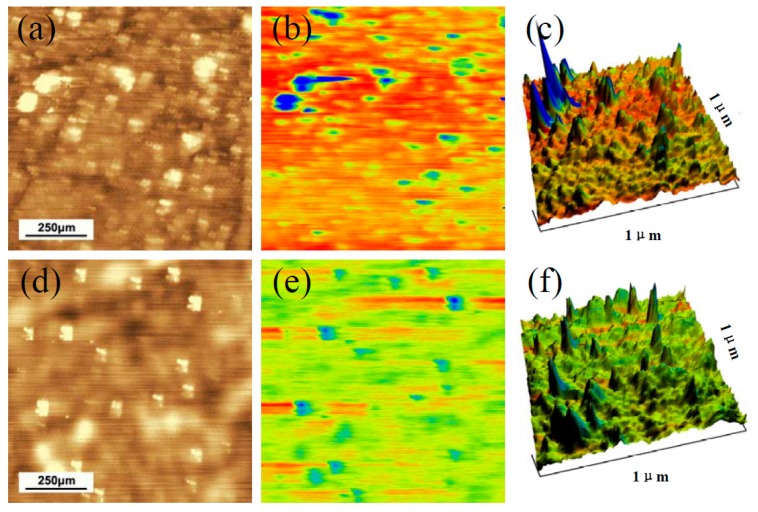
Surface nanoscale morphology for a typical microchannel wall of reduced lead silicate glasses. (**a**,**d**) AFM tomography, (**b**,**e**) KFM surface potential tomography and (**c**,**f**) 3D-mixed tomography.

**Figure 7 materials-12-01183-f007:**
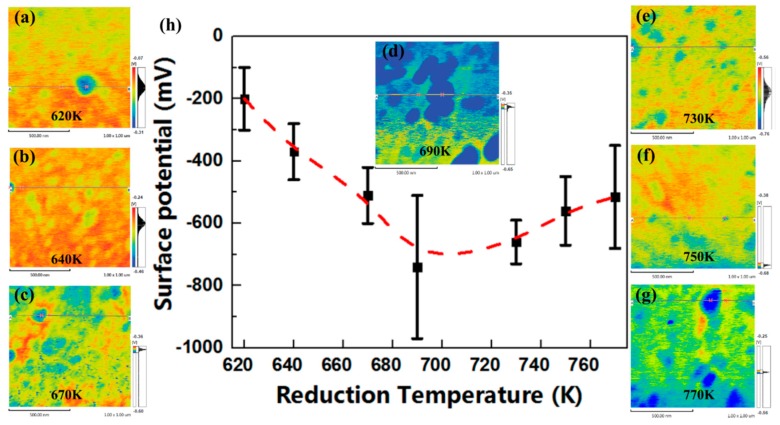
The surface topographies and extrema of CPD of reduced lead silicate glasses at the reduction temperatures from 620 K to 770 K (**a**–**h**).

**Figure 8 materials-12-01183-f008:**
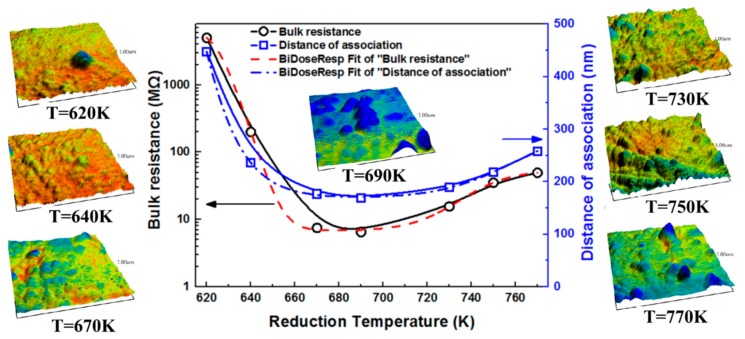
Profiles and fitting charts of the reduction temperature to the distance of [Pb] atom aggregates at the reduced temperature from 620K to 770K. The 3D-mixed tomography of AFM and KFM for silicate glass at each reduced temperature is shown in Figure 7, displaying temperature dependence and nanoparticle-spacing dependence of resistivity of lead silicate glass MCPs.

**Figure 9 materials-12-01183-f009:**
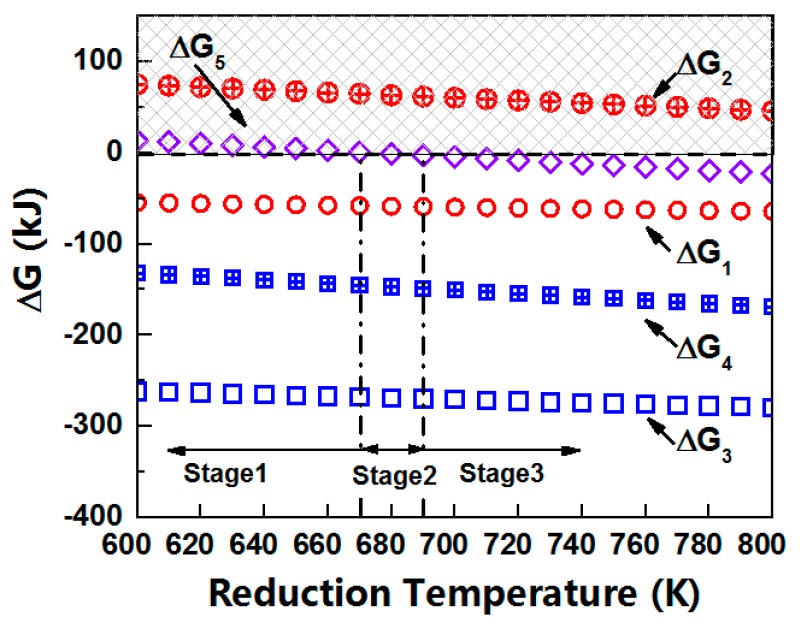
Gibbs free energy changes (ΔG) of the oxidation-reduction reactions in the thermal hydrogen of lead silicate glass on the surface of MCPs from 620 K to 770 K. Only the negative ΔG can react spontaneously at constant pressure (P = 1.1 × 10^5^ Pa) and a stationary temperature.

**Figure 10 materials-12-01183-f010:**
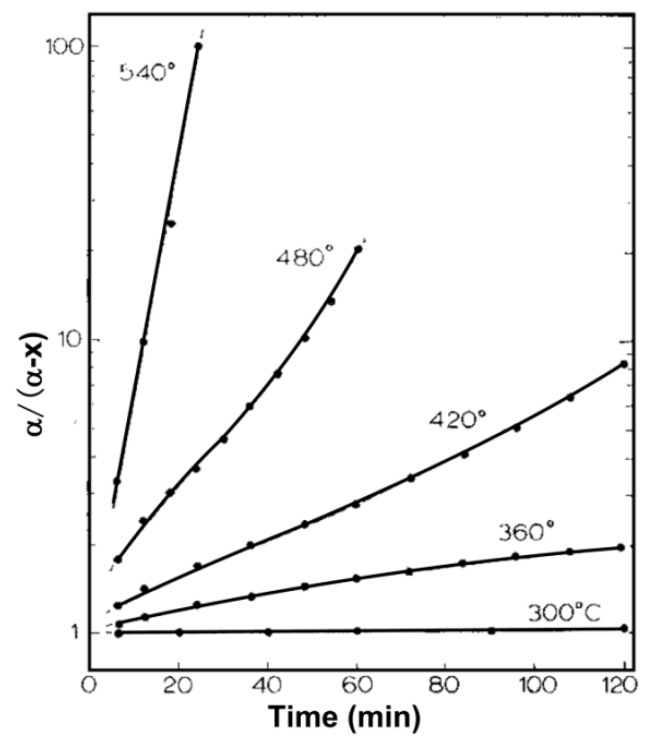
First order kinetic plot of the reduction data for lead oxides [24]. The required reduction time to reach the same content of Pb was shortened greatly when the temperature exceeded 690 K, where α is the fraction of reduction.

**Table 1 materials-12-01183-t001:** The nominal composition of CML865 lead silicate glass (%, mol fraction).

	Pb_3_O_4_	K_2_O	Na_2_O	BaO	MgO	Al_2_O_3_	SiO_2_
CML865	13.6	6.1	2.0	1.2	1.7	1.1	Bal.

**Table 2 materials-12-01183-t002:** The hydrogen reduction parameters of the MCP multipliers.

**No.**	1	2	3	4	5	6	7
**Reduction Temperature/K**	620	640	670	690	730	750	770
**Reduction Time/h**	180						
**Hydrogen Flow/mL·min^−1^**	10						
**Hydrogen Pressure/10^5^ Pa**	1.1						

**Table 3 materials-12-01183-t003:** The fitting data of the XPS spectra Pb 4f for the microchannel wall of CML865 MCP.

			Pb^0^		–Pb–O–		O=Pb=O	
			4f 7/2	4f 5/2	4f 7/2	4f 5/2	4f 7/2	4f 5/2
Before reduction		B.E (eV)	-	-	138.8	143.6	138.2	142.7
		Area%	-	-	35.8	31.1	22.4	10.6
		FWHM (eV)	-	-	1.19	1.11	1.29	0.91
After reduction	T = 620 K	B.E (eV)	136.7	141.6	138.6	143.5	138.2	143.1
		Area%	10.9	9.0	35.2	25.1	11.0	8.8
		FWHM (eV)	1.50	1.52	1.92	1.84	1.55	1.55
	T = 640 K	B.E (eV)	136.8	141.6	138.6	143.5	138.1	142.9
		Area%	14.3	12.6	32.3	24.9	10.1	5.8
		FWHM (eV)	1.65	1.80	1.75	1.72	1.42	1.23
	T = 670 K	B.E (eV)	136.7	141.6	138.6	143.5	138.0	142.9
		Area%	15.1	11.8	36.1	27.5	5.9	3.6
		FWHM (eV)	1.41	1.41	1.87	1.82	1.63	1.22
	T = 690 K	B.E (eV)	136.6	141.5	138.6	143.4	137.7	142.5
		Area%	16.7	14.0	35.9	26.7	4.2	2.6
		FWHM (eV)	1.32	1.38	1.94	1.89	1.67	1.33
	T = 770 K	B.E (eV)	136.6	141.1	138.5	143.4	137.8	142.7
		Area%	16.7	13.2	38.9	29.4	1.3	0.6
		FWHM (eV)	1.54	1.51	1.84	1.82	1.28	0.51

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
