# Peer review of "Reduction Temperature-Dependent Nanoscale Morphological Transformation and Electrical Conductivity of Silicate Glass Microchannel Plate"

_materials, 2019, doi:10.3390/ma12071183_

Round 1
Reviewer 1 Report
The manuscript entiteled "Reduction temperature-dependent nanoscale morphological transformation and electrical conductivity of silicate glass microchannel plate" presents some new results on mechanism of reduction of lead silicate glass under annealing in h2 atmosphere. This results could be interested for technical specialists and scientists wich work with materials for electron multipliers.
But I found some problems on that manuscript.
For first I found that manuscript look not like research article but like laboratory report. I think that the novelity of manuscript is not obvious.
Also there a lot of typos in manuscirpt ,for example in lines 113, 117 etc. Especialy authors should check references in text.
I don't understand why authors talk about PbO2 when they discuss reduction reactions. XPS results don't provide any evidences of presence PbO2 phase.
Author Response
Response to Reviewer 1 Comments
The manuscript entiteled "Reduction temperature-dependent nanoscale morphological transformation and electrical conductivity of silicate glass microchannel plate" presents some new results on mechanism of reduction of lead silicate glass under annealing in h2 atmosphere. This results could be interested for technical specialists and scientists wich work with materials for electron multipliers.
But I found some problems on that manuscript.
Point 1: For first I found that manuscript look not like research article but like laboratory report. I think that the novelity of manuscript is not obvious.
Response 1:Thank you for the critical review and comments. We revised the manuscript and added Highlights.
Highlights:
The influence of reduction temperature on the nanoscale morphological transformation, microscopic potential distributions and electrical conductivity of silicate glass microchannel plate on the mechanism of reductive thermodynamic and kinetic is presented.
A KFM approach was applied to determine the microscopic conductive characteristic of conducting layer on the surface of microchannel wall.
The concentration-dependent bulk resistance mechanism changes from Pb0 concentration- to conductive phase morphology assisted as the increase in reduction temperature.
The bulk resistance and the spacing of [Pb]-islands both has the Bidoseresp trend dependence on the reduction temperature in the range of 620K~770K, three stages of PbO, PbO2 reduction in the lead silicate glass were presented on the mechanism of thermodynamic and kinetic.
Point 2: Also there a lot of typos in manuscirpt ,for example in lines 113, 117 etc. Especialy authors should check references in text.
Response 2: We revised the manuscript and corrected the typos. The citations was improved and unified.
Point 3: I don't understand why authors talk about PbO2 when they discuss reduction reactions. XPS results don't provide any evidences of presence PbO2 phase.
Response 3: Indeed, the original statement of XPS spectra is not correct. We revised the section of 3.2. (XPS spectra of Pb) as follows:
" Pb ion was kept in a divalent state or a tetravalent state before reduction due to the interaction of Pb–O or O=Pb=O (Fig.4a), while some shoulder peaks of Pb4f occurred in reduced samples (Fig.4b~4f). After decomposing Pb4f 7/2 into three peaks, the binding energy of 138.5 eV~138.6eV was associated with a network modifier of Pb2+ 4f 7/2, 137.7eV~138.2eV was associated with a network modifier of Pb4+ 4f7/2, and 136.6 eV~136.8eV should be attributed to Pb0 4f 7/2 in the reduced sample. Some Pb2+ or Pb4+ ions were reduced into Pb. Pb2+, Pb4+ and Pb0 coexisted on the surface of reduced lead silicate glasses. With the hydrogen reduction and the rising reduction temperature, the binding energies of Pb4+ 4f, Pb2+ 4f and Pb0 4f peaks resulted in a tiny negative shift. We believed that the lower electronegativity of Pb0 led an increasing electron density and these decreasing binding energies."
Figure 4. XPS spectra of Pb 4f before and after reduction. (a) before reduction;(b~f) after reduction; (b),(c),(d),(e),(f), Tr=620K, 640K, 670K, 690K and 770k, respectively; the binding energy differences between Pb 4f7/2 and Pb 4f5/2 of Pb0, Pb-O and O=Pb=O are set as 4.86eV in the peak-fit processingaccording to the "handbook of X-ray photoelectron spectroscopy"[22], the [Pb] "shoulder" peaks were observed in the XPS spectrum after reduction.
" It’s worth to note that the peak area ratio of Pb4+ 4f continued to decline by the raising reduction temperature (Fig.5 the gray circles), even was negligible at 770K, shown in Fig.4(e) and Table 2. The MCP glass was generally reduced at 690K~770K for the applicable bulk resistance, the concentrations of Pb4+ 4f 7/2 were almost less than 4% (4.2%~1.3%, as shown in Fig. 4(e) and Fig. 4(f)), so that might account for the neglect of the PbO2 in many investigations into XPS spectra of the reduced lead silicate glass [16,17,21]."
" Meanwhile, as the hydrogen temperature increased, the concentration of Pb0, PbO and PbO2 showed different changing tendency. The concentration of Pb0 4f 7/2 first increased and then reached a level (about 16.7% ) from 620K to 770K, shown in Fig.5 (the purple squares); the content of Pb4+ 4f 7/2 showed a progressive decreasing as the increasing reduction temperature, the decreasing amplitude of which was great in the range of 640K to 690k (almost -140%), while a smaller amplitude in the range of 620K~640K or 690K~770k (almost -60%); the lever of Pb2+ 4f 7/2 was volatile from 32.3% to 38.9% since the rising reduction temperature, as shown in Fig.5 (the blue dots). The concentrations of PbO and Pb0 were not in the inversely proportional relationship, which confirmed the other compounds enabled to react to Pb0 in the silicate glass, such as PbO2. These non-linear rising/declining Pb concentration-hydrogen temperature behavior is attributed to the behaviors of thermodynamics and kinetics of PbO and PbO2 in the hydrogen reduction of lead silicate glass MCPs, which would be discussed in detail in the section of 3.4."
" Moreover, we noticed that the concentration of Pb0 4f 7/2 was kept a constant of about 16.7% at 690K and 770K, where the bulk resistance of MCPs was 6.5MΩ and 50MΩ, respectively, the two had seven times difference. It indicated that the bulk resistance of MCP was not just relevant to the content of Pb0 in the conducting layer of the channel wall, which might also be related to the morphology of conductive phases. To understand this seemingly paradoxical property, we measured the surface morphology and microscopic potential distribution of lead silicate glass MCPs by AFM and KFM."
Figure 5. The concentration of Pb 4f 7/2 in the conducting layer of the channel wall before and after hydrogen reduction. On the left of the break in abscissa was before reduction, on the left of which was after reduction at different reduction temperatures. The purple squares, blue dots and gray circles means the area ratio of 4f 7/2 binding energy peak of Pb0, PbO and PbO2, respectively.
Table 2 The fitting data of XPS spectra Pb 4f for microchannel wall of CML865 MCP
Pb0 | -Pb-O- | O=Pb=O | ||||||
4f 7/2 | 4f 5/2 | 4f 7/2 | 4f 5/2 | 4f 7/2 | 4f 5/2 | |||
Before reduction | B.E (eV) | - | - | 138.8 | 143.6 | 138.2 | 142.7 | |
Area% | - | - | 35.8 | 31.1 | 22.4 | 10.6 | ||
FWHM (eV) | - | - | 1.19 | 1.11 | 1.29 | 0.91 | ||
After reduction | T=620K | B.E (eV) | 136.7 | 141.6 | 138.6 | 143.5 | 138.2 | 143.1 |
Area% | 10.9 | 9.0 | 35.2 | 25.1 | 11.0 | 8.8 | ||
FWHM (eV) | 1.50 | 1.52 | 1.92 | 1.84 | 1.55 | 1.55 | ||
T=640K | B.E (eV) | 136.8 | 141.6 | 138.6 | 143.5 | 138.1 | 142.9 | |
Area% | 14.3 | 12.6 | 32.3 | 24.9 | 10.1 | 5.8 | ||
FWHM (eV) | 1.65 | 1.80 | 1.75 | 1.72 | 1.42 | 1.23 | ||
T=670K | B.E (eV) | 136.7 | 141.6 | 138.6 | 143.5 | 138.0 | 142.9 | |
Area% | 15.1 | 11.8 | 36.1 | 27.5 | 5.9 | 3.6 | ||
FWHM (eV) | 1.41 | 1.41 | 1.87 | 1.82 | 1.63 | 1.22 | ||
T=690K | B.E (eV) | 136.6 | 141.5 | 138.6 | 143.4 | 137.7 | 142.5 | |
Area% | 16.7 | 14.0 | 35.9 | 26.7 | 4.2 | 2.6 | ||
FWHM (eV) | 1.32 | 1.38 | 1.94 | 1.89 | 1.67 | 1.33 | ||
T=770K | B.E (eV) | 136.6 | 141.1 | 138.5 | 143.4 | 137.8 | 142.7 | |
Area% | 16.7 | 13.2 | 38.9 | 29.4 | 1.3 | 0.6 | ||
FWHM (eV) | 1.54 | 1.51 | 1.84 | 1.82 | 1.28 | 0.51 |

Reviewer 2 Report
The manuscript under evaluation is devoted to rather important studies on morphological features of MCP samples. It is well written, organised, with reasonable citation. The research presented is based on well known techniques adequately used by the authors for getting the research aims. The manuscript can be accepted for publication after minor revision, which is mainly due to:
- the mistakes in English to be corrected via careful reading;
- the unclear citations, especially in the introductory part;
- multiple repetition of the extensions for short names of various techniques (AFM, KFM, etc) in the introduction.
As additional option to improve the paper, seems, it would be nice to have very simple description of various techniques utilised at studying MCP samples. It will help the reader to follow all the processes involved in lead silicate cover of channels' surface as well as the change in the bulk resistance and conduction features of such MCPs.
In my opinion, the introduction reveals a lack in description of the importance for MCP to be modulated by variation of its performance.
Once above comments are taken into account the manuscript can be accepted.
Author Response
Response to Reviewer 2 Comments
The manuscript under evaluation is devoted to rather important studies on morphological features of MCP samples. It is well written, organised, with reasonable citation. The research presented is based on well known techniques adequately used by the authors for getting the research aims. The manuscript can be accepted for publication after minor revision, which is mainly due to:
Point 1:- the mistakes in English to be corrected via careful reading;
Response 1: Thank you for the review and comments. We revised the manuscript and corrected the typos.
Point 2:- the unclear citations, especially in the introductory part;
Response 2: We improved and unified the citations in the full text.
Point 3:- multiple repetition of the extensions for short names of various techniques (AFM, KFM, etc) in the introduction.
Response 3: Thank you for the careful review. We revised the manuscript and removed the repetitive extensions for short names of various techniques.
Point 4: As additional option to improve the paper, seems, it would be nice to have very simple description of various techniques utilised at studying MCP samples. It will help the reader to follow all the processes involved in lead silicate cover of channels' surface as well as the change in the bulk resistance and conduction features of such MCPs.
Response 4: We revised the introduction part of the manuscript and the following was added into the text:" The core and clad lead silicate glasses formed the microchannel structure after drawing into fibers, assembling to bundle, fusing, slicing and polishing to wafers, and the wet chemically etching to remove the cores. Then after a hot hydrogen reduction it produced a conducting layer and formed a secondary electron emission layer on the surface of the microchannel wall, lastly, a nichrome layer was evaporated to provide for electrical contact. The hydrogen reduction was one of the most important processes where it possessed a proper electrical conductivity. "
Point 5: In my opinion, the introduction reveals a lack in description of the importance for MCP to be modulated by variation of its performance.
Response 5: We revised the manuscript and added the following into the text:" The bulk resistance of any microchannel plate can determine the application field of MCP multiplier, which influences the dynamic range and voltage tolerance of MCP. The requirements of dynamic range and voltage tolerance for the bulk resistance are exact opposite. Dynamic range is defined as the range of operational linear relationship between input and output currents of MCP multiplier under a fixed external voltage. Voltage tolerance means the maximum working voltage which MCP multiplier can be endured."
" For better applications, therefore, it is important to know exactly how adjusts the bulk resistance to the desired value to establish relative maximum operational electron multiplication dynamic range and best voltage tolerance."
" The electrical conductivity (the bulk resistance) of MCP can be modulated by the composition of glass materials and the hydrogen reduction parameters. The effects of the composition of glass materials on the conductivity of MCP will be discussed in detail in a subsequent paper. In this paper, we focused on the effect of reduction procedure on the bulk resistance of MCP."
Point 6: Once above comments are taken into account the manuscript can be accepted.
Response 6: Thank you for the review and comments.

Round 2
Reviewer 1 Report
Authors provided point by point answers for all reviewers questions. I found this version of manuscript suitable for publication in Materials journal.